# Blood Copper Levels and the Occurrence of Colorectal Cancer in Poland

**DOI:** 10.3390/biomedicines9111628

**Published:** 2021-11-05

**Authors:** Piotr Baszuk, Wojciech Marciniak, Róża Derkacz, Anna Jakubowska, Cezary Cybulski, Jacek Gronwald, Tadeusz Dębniak, Tomasz Huzarski, Katarzyna Białkowska, Sandra Pietrzak, Magdalena Muszyńska, Józef Kładny, Steven A. Narod, Jan Lubiński, Marcin R. Lener

**Affiliations:** 1International Hereditary Cancer Center, Department of Genetics and Pathology, Pomeranian Medical University in Szczecin, ul. Unii Lubelskiej 1, 71-252 Szczecin, Poland; piotr.baszuk@pum.edu.pl (P.B.); wojciech.marciniak@read-gene.com (W.M.); roza.derkacz@gmail.com (R.D.); aniaj@pum.edu.pl (A.J.); cezarycy@pum.edu.pl (C.C.); jgron@pum.edu.pl (J.G.); debniak@pum.edu.pl (T.D.); huzarski@pum.edu.pl (T.H.); katarzyna.kaczm@gmail.com (K.B.); sandra.pietrzak@pum.edu.pl (S.P.); magda_mac@wp.pl (M.M.); lubinski@pum.edu.pl (J.L.); 2Read-Gene, Grzepnica, ul. Alabastrowa 8, 72-003 Grzepnica, Dobra (Szczecińska), Poland; 3Department of Clinical Genetics and Pathology, University of Zielona Góra, ul. Zyty 28, 65-046 Zielona Góra, Poland; 4Department of General Surgery and Surgical Oncology, First Clinical Hospital of Pomeranian Medical University in Szczecin, ul. Unii Lubelskiej 1, 71-252 Szczecin, Poland; jkladny@onet.pl; 5Women’s College Research Institute, Toronto, ON M5G 1N8, Canada; steven.narod@wchospital.ca; 6Dalla Lana School of Public Health, University of Toronto, Toronto, ON M5T 3M7, Canada

**Keywords:** copper, colorectal cancer, biomarkers

## Abstract

There is a need for sensitive and specific biomarkers for the early detection of colorectal cancer. In this retrospective study, we assessed whether a high blood copper level was associated with the presence of colorectal cancer. The blood copper level was measured among 187 colorectal cancer patients and 187 matched controls. Cases and controls were matched for sex, smoking status (yes/no) and year of birth. Among the cases, the mean blood copper level was 1031 µg/L (range 657 µg/L to 2043 µg/L) and among the controls, the mean blood copper level was 864 µg/L (range 589 µg/L to 1433 µg/L). The odds ratio for colorectal cancer for those in the highest quartile of copper level (versus the lowest) was 12.7 (95% CI: 4.98–32.3; *p* < 0.001). Of the patients with stage I–II colon cancer, 62% had a copper level in the highest quartile. A blood copper level in excess of 930 µg/L is associated with an increase in the prevalence of colorectal cancer in the Polish population and its potential use in early detection programs should be considered.

## 1. Introduction

Colorectal cancer (CRC) is one of the most common cancers in men and women [1]. The early diagnosis of colorectal cancer is promoted as a means to reduce the burden of the disease in society. There are several approaches to the early diagnosis of colorectal cancer, including pre-screening of stool samples or blood tests for relevant biomarkers (proteins; DNA; mRNA and long non-coding RNA; microRNA; metabolites and gut microbiome) followed by colonoscopies or sigmoidoscopies where appropriate [2]. Others have recommended computer-assisted algorithms for evaluating complete blood counts (CBCs) or platelets [3,4,5]. To date, non-invasive tests for colorectal cancer pre-screening have not been proven to be effective in terms of reducing cancer mortality.

Potential biomarkers for cancer can be identified by using a case-control design. The biomarkers are measured in newly-diagnosed cancer cases and in closely matched healthy controls and the levels are compared. It is important to acquire the blood sample prior to treatment to avoid the possibility that the treatment influences the level of the biomarker.

Several trace elements are critical for human health [6,7,8]. One essential micronutrient is copper (Cu). It is required as a catalytic cofactor or as a structural component for proteins and plays a role in critical biological functions such as enzyme activity, oxygen transport and cell signaling. Cu is highly redox active, readily donating and accepting electrons to shift between its two valence states (Cu^+^ ⇔ Cu^2+^). Many critical enzymes depend on this activity and hence Cu plays an important role in biological oxidation–reduction (redox) reactions [9]. It can catalyze the production of free radicals and this can be damaging to lipids, proteins, DNA and other biomolecules [10,11]. Cu is not a carcinogen per se, but these activities make it potentially toxic.

In patients with cancer (or inflammation) Cu concentrations rise in the plasma and the synthesis and secretion of ceruloplasmin by the liver is enhanced. Ceruloplasmin is a major Cu transporter that binds Cu in circulation [12]. The elevated ceruloplasmin concentrations in each of these conditions would provide additional Cu uptake by cells in normal tissues and as well as cancer cells. Transcuperine (a second Cu-binding protein) appears to be increased in malignancy. Enhanced uptake of Cu from non-ceruloplasmin fractions of plasma, which together account for the relatively high levels of Cu in tumor cells, was demonstrated in neoplastic cells. Cu also plays a role in angiogenesis, a requirement for tumor growth, which may be mediated by Cu-dependent amine oxidases [13].

Several studies have reported high serum Cu levels in a wide range of cancers including lymphoma, reticulum cell sarcoma, laryngeal carcinomas, cervical, breast, pancreas, stomach and lung cancers [14,15,16]. Elevated serum Cu levels correlated with disease stage in breast and colorectal cancer [17,18].

The aim of the study was to assess whether blood Cu levels are associated with the presence of colorectal cancer.

## 2. Materials and Methods

### 2.1. Study Group

A total of 187 subjects with pathologically-confirmed colorectal cancer were enrolled for study. Cases originated from the Department of General Surgery and Surgical Oncology, Clinical Hospital of Pomeranian Medical University in Szczecin. Blood samples were taken from cases shortly after diagnosis and prior to treatment (between 2011 and 2017).

The controls were identified from a research registry housed at the International Hereditary Cancer Center, Pomeranian Medical University in Szczecin. Controls were participants in a population-based study of the 1.3 million inhabitants of Poland designed to identify familial aggregations of cancers conducted by our center. Blood samples from participants of the aforementioned registry were taken between 2012 and 2017.

Each case was assigned to one control matched for sex, smoking status (yes/no) and year of birth (±3 years).

The study was conducted in accordance with the Helsinki Declaration and with the consent of the Ethics Committee of Pomeranian Medical University in Szczecin under the number KB-0012/73/10 of 21 June 2010. All participants provided written informed consent to be enrolled in the herein study between 2011 and 2017. The overall characteristics of individuals enrolled into the following study are presented in Table 1.

### 2.2. Measurement of Blood Copper Level

For analysis, 10 mL of peripheral blood was collected into a vacutainer tube containing ethylenediaminetetraacetic acid (EDTA) from all study participants and then stored at −80 °C until the day of analysis. Study subjects fasted for at least six hours before sample collection. Determination of ^65^Cu was performed using an inductively coupled plasma mass spectrometer (ICP-MS) (ELAN DRC-e, PerkinElmer, Concord, ON, Canada). Before each assay, the instrument was tuned to achieve the manufacturers’ optimum criteria. Oxygen was used as a reaction gas. The spectrometer was calibrated using an external calibration technique. Calibration standards were prepared fresh daily, from 10 µg/mL Multi-Element Calibration Standard 3 (PerkinElmer Pure Plus, Shelton, CT, USA) by dilution with a blank reagent to the final concentration of 5; 10 and 50 µg/L for Cu determination. Correlation coefficients for calibration curves were always greater than 0.999. Matrix-matched calibration was used. Rhodium was set as the internal standard.

The analysis protocol assumed a 30-fold dilution of blood in a blank reagent. The blank reagent consisted of high purity water (>18 MΩ), TMAH (AlfaAesar, Kandel, Germany), Triton X-100 (PerkinElmer, Shelton, CT, USA), n-butanol (Merck, Darmstadt, Germany), rhodium (PerkinElmer, Shelton, CT, USA), gold (VWR, Steinheim, Germany) and EDTA (Sigma-Aldrich, Leuven, Belgium).

The accuracy and precision of all measurements were tested using certified reference material (CRM), Clincheck Plasmonorm Blood Trace Elements Level 1 (Recipe, Munich, Germany). Recovery rates were between 80% and 105% for analyzed elements; calculated recurrence (Cv%) was below 15% for all the measured elements. The testing laboratory was a member of two independent external quality assessment schemes: LAMP organized by the CDC (LAMP: Lead And Multielement Proficiency Program; CDC: Center for Disease Control) and QMEQAS organized by the Institut National de Santé Publique du Québec (QMEQAS: Quebec Multielement External Quality Assessment Scheme).

### 2.3. Statistical Analysis

All study participants were assigned to one of four quartiles according to increasing Cu concentrations among the controls. The cut-off levels for the quartiles were used in all calculations. To estimate the association of Cu levels with colorectal cancer occurrence, odds ratios (OR) and 95% confidence intervals (CIs) were calculated using univariable conditional logistic regression. The baseline (first quartile) of blood Cu was taken as the reference group. All statistical calculations were performed using: R: A language and environment for statistical computing. R Foundation for Statistical Computing, Vienna, Austria; https://www.R-project.org/ (R version 4.04; accessed on 10 October 2020).

## 3. Results

Among the 187 controls, the mean blood Cu level was 864 µg/L (range 589 µg/L to 1433 µg/L) and among all cases, the mean blood Cu level was 1031 µg/L (range 657 µg/L to 2043 µg/L). In the overall analysis there was a positive association between blood Cu level and colorectal cancer (Table 2). The odds ratio for those within the highest Cu quartile, compared to the baseline quartile was 12.7 (95% CI: 4.98–32.3; *p* < 0.001).

Among all women, the mean blood Cu level was 1003 µg/L (range 657 µg/L to 1899 µg/L) and among all men, the mean blood Cu level was 902 µg/L (range 589 µg/L to 2043 µg/L) (*p* < 0.001). The association between Cu level and colon cancer was present for women (Table 3) and for men (Table 4). The odds ratio for a Cu in the highest quartile was stronger in men (OR: 15.1; 95% CI: 4.84–47.2; *p* < 0.001) than in women (OR: 5.25; 95% CI: 1.02–27.0; *p* = 0.047). However, 77% of cases in women and 51% of cases in men had a Cu level in the highest quartile.

Cu levels according to cancer characteristics are presented in Table 5. Statistically significant higher mean Cu levels were found for cancer stage III–IV compared to stage I–II (*p* < 0.04) and G3 cancers compared with G1–G2 tumors (*p* < 0.001). There was no difference in metastases or cancer localization.

The probability of cancer occurrence depending on Cu levels after stratification of individuals into quartiles is presented in Table 6, Table 7, Table 8, Table 9, Table 10, Table 11, Table 12 and Table 13.

An association was present both for patients with stage I–II colorectal cancer (Table 6) and stage III–IV colorectal cancer (Table 7).

An association was present also for patients with low-grade colon cancer (Table 8) and with high-grade colon cancer (Table 9).

The highest levels of Cu associated with increased frequency of cancer were found irrespective of the occurrence of metastases (Table 10 and Table 11).

The copper–cancer association also shows similar characteristics in different cancer localizations (Table 12 and Table 13).

## 4. Discussion

In this retrospective study, we found a strong and significant association between blood Cu level and the occurrence of colorectal cancer in the Polish population. The odds ratio for the highest quartile of blood Cu, compared to the baseline quartile was 12.7 (95% CI: 4.98–32.3; *p* < 0.001). The association was strong for cancers detected at the early stage as well as at late stage and was present irrespective also of grading, metastases and cancer localization.

The odds ratio according to highest quartile was stronger in men than in women—OR: 15.1 vs. 5.25 respectively. There is evidence that blood Cu level in women is dependent on the use of hormones; possibly the higher blood Cu levels in women are caused by the hormone replacement therapy that is in use by 50% of Polish women above age 50 years [19,20].

Several prior studies have evaluated the association between Cu and colorectal cancer. Table 14 and Table 15 summarize these studies. In these studies the research samples were either collected before the onset of the disease (prospective studies) or after diagnosis (case-control studies).

Several previous studies reported positive associations between serum Cu level and colorectal cancer risk or occurrence. The largest study (the EPIC study) identified 966 incidence cases of colorectal cancer in the cohort. These were matched with 966 healthy controls. Blood was taken up to six years prior to diagnosis. Among those who developed colorectal cancer within two years of the blood draw, the odds ratio for a Cu level in the highest quintile compared to the lowest quintile was 4.00 (95% CI: 1.74–9.16). Among those who developed colorectal cancer more than two years after the blood draw, the odds ratio for a Cu level in the highest quintile compared to the lowest quintile was 1.04 (95% CI: 0.69–1.56) [21]. These data suggest that an elevated serum Cu is a biomarker for the presence of existing colorectal cancer rather than a risk factor for the development of cancer. In our study, the blood was taken at the time of diagnosis and the odds ratios were more extreme than those in the EPIC study. It is not clear to what extent the association is present or attenuated if the blood had been taken two years in the past.

In the second prospective study, NHANES there was an association seen between higher Cu level and increased risk of colorectal cancer (OR: 1.71; 95% CI: 0.37–7.88) but this was not statistically significant (*p* = 0.4) [22]. However, there were only 24 cases in the NHANES study and the data were not restricted to those diagnosed shortly after the blood draw.

A significant association between high Cu level and occurrence of colorectal cancer has been reported in other case control studies [18,23,24,25,26], but these were much smaller in size (Table 15). There are two small studies with results in contrast to our observations. A significantly lower (<0.001) serum Cu level in CRC patient groups (0.47 mg/L) was observed in a retrospective study of 90 CRC patients and 30 healthy persons (mean serum Cu level: 0.80 mg/L) from Iraq [25]. In a small case-control study from the Moravian region of the Czech Republic, conducted on 17 CRC patients and 7 control subjects, a higher (although not significantly) serum Cu level was found among controls (Cu level 0.95 mg/L vs 1.21 mg/L respectively; *p* = 0.899) [26].

Our data indicate that there are systematic differences in blood Cu levels between colorectal cancer patients and unaffected individuals in the Polish population. A blood Cu level above 900 µg/L was found in 132 of 187 cancer patients (sensitivity = 71%; specificity = 64%) and above 1000 µg/L is detecting 87 of 187 cancer patients (sensitivity = 47%; specificity = 88%).

We reported previously that serum selenium level is also a marker for the occurrence of colorectal cancer [27], however with lower sensitivity and specificity than Cu.

The mechanism(s) that cause Cu levels to increase in the blood/serum of cancer patients is (are) not known. In a mouse model of carcinoma, the occurrence of elevated serum Cu was found to be concomitant with a decrease in Cu within the liver. This suggests that Cu distribution around the body, which is mediated by the liver, may be fundamentally altered by cancer [9,28]. There is still no information on whether cellular transformation to malignancy can drive Cu accumulation, or on the mechanisms by which cells adapt to tolerate the ensuing oxidative pressure [9]. In our case-control study, the Cu measurements were made after the diagnosis of colorectal cancer, although before therapy.

The strengths of our study include a large number of patients from the same geographical region who were diagnosed in the same institution and the blood samples in the cases that were taken shortly after diagnosis but before treatment. A limitation of our study is the relatively small sample size, but the observed associations for Cu levels were strong consistent and highly significant. We did not have information on the use of hormone replacement therapy. The level of blood Cu in the population may vary between countries and the laboratory assays used may also show differences. Therefore it is important to establish norms in each population if blood Cu is to be used as a clinical test.

Our study suggests that blood Cu level may be a marker of colorectal cancer and may be informative in selecting individuals for colonoscopy who might benefit from surveillance for early detection of colorectal cancer. At present, the choice of biomarkers useful for population screening for colonoscopies is very limited. The FIT (fecal immunochemical test) is the most frequently used for screening. The Cologuard^®^ stool test with complex targets or Epi ProColon test analyzing SEPT9 methylation in the blood, is prohibitively expensive (USD 600). However, FIT sensitivity is relatively modest (66–74%) [2]. Thus, further research incorporating panels of various biomarkers such as Cu levels might increase sensitivity and specificity.

## 5. Conclusions

A high blood Cu level (>900 µg/L) is associated with a significantly increased occurrence of colorectal cancer in the Polish population. Determination of the level of Cu has the potential as a marker for the selection of patients for further surveillance with colonoscopy.

## Figures and Tables

**Table 1 biomedicines-09-01628-t001:** Characteristics of individuals enrolled into the study.

Characteristic	Cases, *n* = 187	Controls, *n* = 187
Year of birth	1923–1985 (1949) ^1^	1926–1986 (1949)
Age at blood draw	32–94 (68)	31–91 (68)
Year of blood draw	2011–2017 (2014)	2012–2019 (2014)
Sex		
Female	83 (44%) ^2^	83 (44%)
Male	104 (56%)	104 (56%)
Smoking		
No	141 (75%)	140 (75%)
Yes	46 (25%)	47 (25%)
Colon cancer stage		
I	27 (14%)	-
II	74 (40%)	-
III	57 (30%)	-
IV	14 (7.5%)	-
Missing	15 (8.0%)	-
Site		
Colon	81 (43%)	-
Rectum/Rectosigmoid Junction	91 (49%)	-
Missing	15 (8%)	-
Grading		
G1	14 (7.5%)	-
G2	109 (58%)	-
G3	33 (18%)	-
Mucinous	10 (5.3%)	-
Missing	21 (11%)	-
Metastases present		-
No	100 (53%)	-
Yes	69 (37%)	-
Missing	18 (9.6%)	-

^1^ Range (Mean); ^2^
*n* (%).

**Table 2 biomedicines-09-01628-t002:** Blood copper levels and CRC occurrence.

Copper Level	Cases *n* = 187	Controls *n* = 187	OR	95% CI	*p*-Value
Q1 589–768 µg/L	11 (5.9%)	47 (25%)	—	—	
Q2 768–854 µg/L	33 (18%)	46 (25%)	3.34	1.30–8.61	0.012
Q3 855–930 µg/L	26 (14%)	47 (25%)	3.46	1.25–9.56	0.017
Q4 931–2043 µg/L	117 (63%)	47 (25%)	12.7	4.98–32.3	<0.001

**Table 3 biomedicines-09-01628-t003:** Blood copper levels and the occurrence of colorectal cancer among women.

Copper Level	Cases *n* = 83	Controls *n* = 83	OR	95% CI	*p*-Value
Q1 589–768 µg/L	2 (2.4%)	8 (9.6%)	—	—	
Q2 768–854 µg/L	5 (6.0%)	15 (18%)	0.78	0.11–5.40	0.8
Q3 855–930 µg/L	12 (14%)	25 (30%)	1.17	0.18–7.62	0.9
Q4 931–2043 µg/L	64 (77%)	35 (42%)	5.25	1.02–27.0	0.047

**Table 4 biomedicines-09-01628-t004:** Blood copper levels and the occurrence of colorectal cancer among men.

Copper Level	Cases *n* = 104	Controls *n* = 104	OR	95% CI	*p*-Value
Q1 589–768 µg/L	9 (8.7%)	39 (38%)	—	—	
Q2 768–854 µg/L	28 (27%)	31 (30%)	4.94	1.59–15.3	0.006
Q3 855–930 µg/L	14 (13%)	22 (21%)	4.56	1.31–15.8	0.017
Q4 931–2043 µg/L	53 (51%)	12 (12%)	15.1	4.84–47.2	<0.001

**Table 5 biomedicines-09-01628-t005:** Copper levels by colon cancer characteristics.

Characteristics		*n*	Mean Copper Level, µg/L (Range)	*p*-Value
Stage	I–II	101	1002.7 (663.1–1899.3)	0.04
	III-IV	71	1083.7 (657.1–2043.2)	
Grade	G1–G2	123	1000.7 (663.1–1899.3)	<0.001
	G3	33	1149.5 (791.6–2043.2)	
Metastases	No	100	1016.9 (663.1–1899.3)	0.2
	Yes	69	1069.2 (657.1–2043.2)	
Localisation	Colon	81	1048.7 (701.8–1548.7)	0.33
	Rectum/rectosigmoid junction	91	1027.1 (657.1–2043.2)	

**Table 6 biomedicines-09-01628-t006:** Blood copper levels and the occurrence of stage I–II colorectal cancer.

Copper Level	Cases *n* = 101	Controls *n* = 101	OR	95% CI	*p*-Value
Q1 589–768 µg/L	4 (4.0%)	25 (25%)	—	—	
Q2 768–854 µg/L	21 (21%)	25 (25%)	7.11	1.52–33.3	0.013
Q3 855–930 µg/L	13 (13%)	26 (26%)	5.59	1.12–28.0	0.036
Q4 931–2043 µg/L	63 (62%)	25 (25%)	21.0	4.54–97.0	<0.001

**Table 7 biomedicines-09-01628-t007:** Blood copper levels and the occurrence of stage III–IV colorectal cancer.

Copper Level	Cases *n* = 71	Controls *n* = 71	OR	95% CI	*p*-Value
Q1 589–768 µg/L	3 (4.2%)	17 (24%)	—	—	
Q2 768–854 µg/L	10 (14%)	18 (25%)	1.70	0.38–7.56	0.5
Q3 855–930 µg/L	11 (15%)	16 (23%)	2.59	0.53–12.7	0.2
Q4 931–2043 µg/L	47 (66%)	20 (28%)	7.99	2.15–29.7	0.002

**Table 8 biomedicines-09-01628-t008:** Blood copper levels and the occurrence of colorectal cancer among G1-G2 patients.

Copper Level	Cases *n* = 123	Controls *n* = 123	OR	95% CI	*p*-Value
Q1 589–768 µg/L	7 (5.7%)	29 (24%)	—	—	
Q2 768–854 µg/L	25 (20%)	33 (27%)	4.19	1.15–15.2	0.030
Q3 855–930 µg/L	20 (16%)	30 (24%)	4.76	1.21–18.7	0.026
Q4 931–2043 µg/L	71 (58%)	31 (25%)	13.6	3.80–48.4	<0.001

**Table 9 biomedicines-09-01628-t009:** Blood copper levels and the occurrence of colorectal cancer among G3 patients.

Copper Level	Cases *n* = 33	Controls *n* = 33	OR	95% CI	*p*-Value
Q1 589–768 µg/L	0 (0%)	9 (27%)	NA	NA	NA
Q2 768–854 µg/L	1 (3.0%)	7 (21%)	-	-	
Q3 855–930 µg/L	3 (9.1%)	10 (30%)	2.10	0.21–47.5	0.6
Q4 931–2043 µg/L	29 (88%)	7 (21%)	29.0	4.24–591	0.003

**Table 10 biomedicines-09-01628-t010:** Blood copper levels and the occurrence of colorectal cancer among patients with metastases.

Copper Level	Cases *n* = 69	Controls *n* = 69	OR	95% CI	*p*-Value
Q1 589–768 µg/L	2 (2.9%)	16 (23%)	—	—	
Q2 768–854 µg/L	10 (14%)	18 (26%)	2.64	0.50–14.0	0.3
Q3 855–930 µg/L	13 (19%)	16 (23%)	4.87	0.83–28.7	0.080
Q4 931–2043 µg/L	44 (64%)	19 (28%)	11.8	2.47–56.1	0.002

**Table 11 biomedicines-09-01628-t011:** Blood copper levels and the occurrence of colorectal cancer among patients without metastases.

Copper Level	Cases *n* = 100	Controls *n* = 100	OR	95% CI	*p*-Value
Q1 589–768 µg/L	5 (5.0%)	24 (24%)	—	—	
Q2 768–854 µg/L	18 (18%)	26 (26%)	5.41	1.11–26.4	0.037
Q3 855–930 µg/L	12 (12%)	27 (27%)	4.44	0.88–22.5	0.072
Q4 931–2043 µg/L	65 (65%)	23 (23%)	21.4	4.52–101	<0.001

**Table 12 biomedicines-09-01628-t012:** Blood copper levels and the occurrence of cancer localized in the colon.

Copper Level	Cases *n* = 81	Controls *n* = 81	OR	95% CI	*p*-Value
Q1 589–768 µg/L	1 (1.2%)	20 (25%)	—	—	
Q2 768–854 µg/L	17 (21%)	27 (33%)	7.55	0.93–61.2	0.058
Q3 855–930 µg/L	9 (11%)	18 (22%)	10.0	1.08–93.4	0.042
Q4 931–2043 µg/L	54 (67%)	16 (20%)	45.1	5.29–386	<0.001

**Table 13 biomedicines-09-01628-t013:** Blood copper levels and the occurrence of cancer localized in rectum/rectosigmoid junction.

Copper Level	Cases *n* = 91	Controls *n* = 91	OR	95% CI	*p*-Value
Q1 589–768 µg/L	6 (6.6%)	22 (24%)	—	—	
Q2 768–854 µg/L	13 (14%)	17 (19%)	3.27	0.83–13.0	0.091
Q3 855–930 µg/L	16 (18%)	25 (27%)	3.42	0.82–14.2	0.092
Q4 931–2043 µg/L	56 (62%)	27 (30%)	9.89	2.68–36.5	<0.001

**Table 14 biomedicines-09-01628-t014:** Prospective studies on association between copper level and risk of colorectal cancer.

Country	*n*	Cases (*n*)	Controls (*n*)	Material	Time of Material Collection	Mean Cu Level		OR	*p*	Ref.
						Cases	Controls			
10 European countries (Denmark, France, Germany, Greece, Italy, The Netherlands, Norway, Spain, Sweden, U.K.)	1932	966	966	serum	3.8 years (mean 2.1) before CRC diagnosis	1.39 mg/L	1.36 mg/L	1.50	0.02	Stępień, EPIC 2017 [21]
U.S.	4663	24	4639	serum	up to 2 years before diagnosis	1.24 mg/L	1.19 mg/L	1.71	0.429	Zhang, NHANES 2021 [22]

**Table 15 biomedicines-09-01628-t015:** Case-control studies on the association between copper level and occurrence of colorectal cancer.

Country	*n*	Cases (*n*)	Controls (*n*)	Material	Time of Material Collection	Mean Cu Level		*p*	Ref.
						Cases	Controls		
Iran	80	40	40	blood	at diagnosis	0.82 mg/L	0.61 mg/L	0.03	Ranjbary 2020 [23]
Brasil	74	46	28	plasma	at diagnosis	1.2 mg/L	1.06 mg/L	<0.01	Ribeiro 2016 [24]
India	60	30	30	serum	at diagnosis	1.66 mg/L	0.99 mg/L	<0.001	Gupta 1993 [18]
Iraq	120	90	30	serum	at diagnosis	0.47 mg/L	0.80 mg/L	<0.001	Al-Ansari 2020 [25]
Czech Republic	24	17	7	serum	at diagnosis	0.95 mg/L	1.21 mg/L	0.899	Milde 2001 [26]
Poland	374	187	187	blood	at diagnosis	1.03 mg/L	0.86 mg/L	<0.001	Baszuk 2021

## Data Availability

The data presented in this study are available on request from the corresponding author.

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
