# Peer review of "Blood Copper Levels and the Occurrence of Colorectal Cancer in Poland"

_biomedicines, 2021, doi:10.3390/biomedicines9111628_

Round 1

Reviewer 1 Report

The authors Baszuk P. et al., described that blood copper level is higher in colorectal patients than controls and may be a biomarker for the detection of colorectal cancer. This article is interesting and clinically important.

Although the approach is interesting, there some major concerns that should be addressed by the authors.

  • In Materials and Methods or Results section, Authors should explain the characteristics of colorectal cancer (e.g. differentiation of cancer (well, moderate, poorly, …), presence of gene mutation (RAS, BRAF, …), the cite of cancer origin, the organs of distant metastasis).
  • In the result section, please examine the relation between copper level and factors (e.g. differentiation, gene mutation, the cancer origin cite, the organs of metastasis, …) because I think that the causes of higher copper level in colorectal cancer can be explained more clearly.
  • In the Discussion section, Authors should explain that the result do not become constant in other studies. Results were different because of the difference of patient’s characteristics?

Author Response

Dear Reviewer,

We appreciate very much your effort in reviewing our manuscript: „Blood Copper Levels and the Occurrence of Colorectal Cancer in Poland”.

We will try to make all required improvements although we have one problem difficult to solve. In our country NRAS, KRAS and BRAF mutations are examined routinely only since 2 years. Our series of 187 cases is older.

Therefore, to study these mutations, we need to find the blocks in the archives and perform the mentioned molecular analysis.

So we need ~3 weeks and it will cost around 20,000 euros.

Could you please forward our article without analysing metioned mutations?

Sincerely,

Marcin Lener

 and co-authors

Reviewer 2 Report

The work is simple, clear and well presented. The study cohort is small (187 patients vs 187 healthy donors), but it is from same geographical region. Then, I think that this work could be useful for the scientific community

Author Response

Thank you very much for the review.

Round 2

Reviewer 1 Report

  • In Materials and Methods or Results section, please describe the characteristics of colorectal cancer (e.g. differentiation of cancer (well, moderate, poorly, …), the cite of cancer origin, the organs of distant metastasis).
  • In the result section, please examine the relation between copper level and factors (e.g. differentiation, the cancer origin cite, the organs of metastasis, …) because I think that the causes of higher copper level in colorectal cancer can be explained more clearly.

Author Response

Dear Reviewer,

We appreciate very much your effort in reviewing our manuscript: „Blood Copper Levels and the Occurrence of Colorectal Cancer in Poland”.

Please note:

  1. we added characteristics of colorectal cancer (staging, grading, metastases, localization) and we examined the relation between copper level and above factors
  2. English was improved by Prof. S. Narod

Sincerely,

Marcin Lener

 and co-authors

Round 3

Reviewer 1 Report

I think that authors had revised enough.